# Total Health Expenditure and Its Driving Factors in China: A Gray Theory Analysis

**DOI:** 10.3390/healthcare9020207

**Published:** 2021-02-14

**Authors:** Huanhuan Jia, Hairui Jiang, Jianxing Yu, Jingru Zhang, Peng Cao, Xihe Yu

**Affiliations:** School of Public Health, Jilin University, Changchun 130000, Jilin Province, China; hhjia20@mails.jlu.edu.cn (H.J.); jianghairuii@163.com (H.J.); yjxjlu@163.com (J.Y.); 18834184124@163.com (J.Z.); cppengcao@163.com (P.C.)

**Keywords:** health expenditure, socioeconomic factors, predictions and projections, demography, public policy

## Abstract

The continuous growth in total health expenditure (THE) has become a social issue of common concern in most countries. In China, the total health expenditure (THE) is maintaining a rapid growth trend that is higher than that of the economy, which has become increasingly obvious in the 21st century and has brought a heavy burden to the government and residents. To analyze the main driving factors of THE in China in the 21st century and establish a predictive model, gray system theory was employed to explore the correlation degree between THE and nine hot topics in the areas of the economy, population, health service utilization, and policy using national data from 2000 to 2018. Additionally, a New Structure of the Multivariate Gray Prediction Model of THE was established and compared with the traditional grey model and widely used BP neural network to evaluate the prediction effectiveness of the model. We concluded that the Chinese government and society have played a crucial role in reducing residents’ medical burden. Besides this, the improved economy and aging population have increased the demand for health services, leading to the continual increase in THE. Lastly, the improved NSGM(1,N) model achieved good prediction accuracy and has unique advantages in simulating and predicting THE, which can provide a basis for policy formulation.

## 1. Introduction

Across economic development and healthcare settings, it is increasingly recognized that improving living and health standards is important, and improving health is a growing concern. At the same time, the continuous growth in total health expenditure (THE) and the associated economic burden, as an internationally recognized indicator, have become social issues of common concern in most countries [1,2,3,4], reflecting countries’ investment and burden in the health field from a society-wide perspective. According to the most recent data from the Organization for Economic Cooperation and Development (OECD), at the beginning of the 21st century, the proportion of health expenditure of the gross domestic product (GDP) of its member states rose from 7.0% in 2000 to 8.8% in 2019, and the per capita health expenditure in its member states also increased rapidly. For example, the proportion of health expenditure in US GDP rose by 4.42% to 17.0%, ranking first in the world, and per capita medical and health expenditure increased by 142.95% to USD 11,071 [5]. However, Fredell MN [4] Pointed out that despite spending approximately 18% of GDP—more than USD 3.2 trillion—on healthcare (vs. 6–12% in other developed countries), the United States ranks poorly in terms of objective healthcare measures. In another large economic community, the European Union, THE has been increasing sharply over the past two to three decades. On the one hand, THE more than doubled in real terms between 1995 and 2010, and on the other hand, it is still increasing along a continuous and rather stable trend line [6]. Therefore, how to control unreasonable increases in health expenses is an important issue that urgently needs a solution. In this respect, it is necessary to better understand the main driving factors of growth and establish predictive models to grasp the trend of changes in THE so that governments can identify areas for future intervention.

Research on THE is extensive, and the research methods vary. The main driving factors are demographics [7], economics [6,8], and disease [9]. Scholars [10,11] have also analyzed the relationships between education and health expenditure, air quality and health expenditure, and environment and health expenditure. Because of fundamental differences in the health systems, economic levels, population health, ideologies, cultures and regional environments of different countries, and large disparities in the size and growth of THE, the influencing factors of THE and the extent of their influence also differ. Additionally, no standard approach exists for the measurement of the driving factors; thus, the selection and definition of those factors have inevitably been somewhat subjective and dependent on the data available. Therefore, scholars have often selected driving factors according to the characteristics or hot issues of the study area. Previous studies [8,12] have employed instrumental variable quantile regression or generalized estimating equation methods for panel models to analyze THE, and other scholars [13,14] have used logistic regression, boosted decision trees, neural networks, and the ARIMA model to predict THE. However, a common point is that the amount of data used is large and the calculations are complicated, providing no benefits for short-term analyses or situations where there is “poor information”.

Owing to China’s socialist system and large population, the results of previous studies have only reference significance and no decisive significance. In China, THE has grown considerably since economic reform started in 1978, and its growth rate has exceeded that of GDP [15]. This phenomenon has become more obvious in the 21st century, placing a heavy burden on the government and residents. In 2009, due to the excessive increase in medical expenses, the Chinese government began to implement the new health system reform, with one of the main tasks being reducing the burden of medical treatment for residents and alleviating the “difficulty and high cost of getting medical treatment” [16]. However, THE and per capita health expenditure continued to increase rapidly—the average annual growth rate of THE in 2009–2018 was 14.45%, which was higher than the average annual growth rate of GDP (11.12%). The elasticity of health consumption during this period was 1.30; that is, for every 1% increase in GDP, THE increased by 1.30%, and THE accounted for 6.57% of GDP in 2018 [17]. Zhang et al. [9], experts from the China National Health Development Research Center, determined that the elasticity of health consumption is approximately 1.2, which can guarantee the economic sustainability of health financing. After analysis of historical changes in THE in China, and with reference to changes in health financing development trends and the proportion of THE in the GDPs of the OECD countries, 8% of GDP was determined to be the upper limit or warning value of the sustainability of THE in China. However, the growth of THE has been rapid, and if this trend of excessive growth is not controlled in the future in China, it may exceed social and economic affordability, and the sustainability of health funding will not be guaranteed.

Over the first 19 years of the 21st century, not enough information has yet accumulated to analyze the driving factors of a country’s THE and establish a predictive model. Additionally, the growth of health expenditure is affected by objective and subjective factors, the connotations and extensions are difficult to measure and the characteristics are neither obvious nor easy to analyze. However, Deng’s gray system theory, especially the gray relational analysis and gray prediction model, can be used to model, analyze, monitor, and control uncertain systems and solve the problem of uncertain gray information. This theory has been widely used in the economic [18], biological [19] and environmental fields [20], but few studies have applied it in the field of health. Therefore, the correlation analysis and the improved prediction model of gray system theory were employed in the field of health economics in this paper to analyze the main driving factors of the increase in China’s THE since the beginning of the 21st century, and establish a prediction model. At the same time, the traditional BP neural network model was used to determine the accuracy of the prediction model.

## 2. Materials

As mentioned above, the characteristics of each country are different, so the study of Chinese health expenditure cannot completely adopt the variables used by other scholars, although we explored the research in other parts of the world. At the same time, we identified the popular topics of Chinese scholars’ research regarding China’s medical and health system reform. To a certain extent, the more influencing factors are selected, the more accurate the description and prediction of health expenditure will be. The model needs to be not only accurate, but also concise, so we selected only representative variables from the popular topics. Finally, we selected 9 representative factors, including factors in the fields of economy, population, health institutions, and public policy. These data come from the China Statistical Yearbook and China National Health Accounts Report, and the driving factors are described below.

### 2.1. Demographics

After the availability of data, practical possibilities and interrelationships among the factors were considered, and the demographic factors selected for this paper were population growth and aging. The growth of the population has increased the number of potential users of medical services, which will definitely effect change in THE. Additionally, not only the size but also the age structure of the population are critical factors that affect THE. With the increase in average life expectancy and the decrease in birth and death rates, the number and proportion of the elderly population continue to increase, and the problem of aging populations has intensified throughout the world. Aging is also associated with higher risks of chronic diseases, mild disability, cognitive decline, etc. [21]. The prevalence of multimorbidity increases substantially with age and is present in most people aged 65 years and older [22]. In China, by the end of 2018, the number of citizens aged 65 and over had reached nearly 167 million, accounting for 11.94% of the total population, and the increase in elderly people in China has increased the burden on society in terms of chronic diseases contracted by the elderly [23]. A WHO study [24] showed that between 2020 and 2030, the number of elderly individuals suffering from one or more chronic diseases in China is projected to increase by at least 40%, and the proportion of the elderly population will reach 28% by 2040. Therefore, with the increased elderly population, the challenges for medical services and social policy will increase. Scholars [7,25,26] worldwide have investigated the aging of the population as an important driving factor when researching changes in THE. Therefore, the population and the number of people aged 65 years and over were selected as population factors in relation to the change in THE.

### 2.2. Economy

The relationship between GDP and THE is a popular topic in many studies on health economics [27,28]. GDP is an overall economic indicator that measures a country’s total income and can reflect its economic strength. On the one hand, economic development provides more funds for the development of the health field, and the proportion of THE in GDP is an important indicator of a country’s health input; on the other hand, the continuous growth of social economy leads to the improvement of residents’ income level, and the demand for residents’ health services increases, which in turn leads to the increase in THE. Therefore, the development of the national economy is an important factor to promote the growth of THE. Besides this, as China’s economic strength has increased, the living standards of residents have also continuously improved, and the level of national living standards depends on the level of consumption. If residents do not consume or have no money to consume, they cannot benefit from economic growth, and national economic growth will lose its meaning. At the same time, consumption is an important behavior and process in human social and economic activities, because it is not only the end point but also the starting point of economic activities. Finally, China is the most populous country in the world, and the changes in residents’ consumption levels play an important supporting role in economic development. Therefore, household consumption expenditure (HCE) plays a decisive and vital role in the national standard of living and national economy. In this study, GDP and HCE were selected to reflect the relationship between economic factors and the growth of THE.

### 2.3. Public Policy

Policies, including medical, medical insurance, and drug policies, are very complex and difficult to analyze, so a comprehensive analysis of the impacts of various policies on health expenditure is difficult to achieve. However, standardizing financing and compensation mechanisms, improving the health of the population, and sharing the economic risks of disease are the ultimate goals of various policies. Therefore, indicators of the results of health policies were chosen to reflect the relationship between health policies and health expenditure, which is more intuitive and easier to understand. First, according to the International Classification for Health (ICHA), general government health expenditure (GGHE) reflects the role played by governments at all levels and social security funds as fundraisers. The government can affect a country’s health sectors, and subsequently its health outcomes, in several ways, such as the provision of public health services and the coverage of medical services [29]. Moreover, under the Chinese medical and health system, the government plays an irreplaceable leading role in the health service market, providing public goods, improving income distribution, and promoting social equity, so its health investment deeply reflects its emphasis on healthcare and livelihood issues, such as residents’ health and medical burden. Additionally, the social medical security fund has an impact on health service utilization and expenditure [30] that reflects the contribution of various social medical security systems, such as urban and rural medical insurance, urban workers’ medical insurance, new rural cooperative medical insurance and enterprise employee medical and health expenses. At the same time, the new medical reform policies also focus on clarifying government responsibilities and establishing a scientific and effective medical insurance system, so GGHE was selected in this paper to reflect the role of the government and social security departments in the health field. Secondly, out-of-pocket expenditure (OOP) was selected to reflect the economic burden of residents, that is, the cash payment residents must make when receiving various medical and health services, which is also a component of private expenditure on health (PHE) in the ICHA. The size of OOP also affects residents’ access to and choice of medical and health services. Finally, infant mortality (INF) is an important indicator of a country’s health that is associated with a variety of factors, such as maternal health, quality of service, access to medical care, socioeconomic conditions, and public health practices [31,32,33]. Therefore, it was selected to reflect the national health situation.

### 2.4. Health Institutions

Health institutions are the main providers of medical and health services and are responsible for disease diagnosis and treatment. From the perspective of providers of health services, the development of health institutions has improved the accessibility and equity of health services, affecting the provision and utilization of health services. Especially after the new health system reform in China in 2009, the number and scale of health institutions have both increased significantly. However, the development of health institutions provides more space for the occurrence of health expenditures, which is an important factor leading to the continuous increase in THE. Beds and health technicians are the core resource elements of the medical and health service system, and the important indicators of the size of the institution. Therefore, the numbers of beds and health technicians were selected to reflect the impact of the development of health institutions on THE.

## 3. Methods

### 3.1. Gray Correlation Analysis

Gray correlation analysis, which is used to evaluate the main driving factors of THE, measures the degree of correlation among factors in a system based on the similarity or dissimilarity of the development trends. The comparative analysis of factors in the system includes the geometric shapes of several curves, and when the shapes are approximate, the degree of correlation among the factors is significant and the degree of similarity among the objects considerable. Additionally, gray correlation analysis does not require many samples, the typical distribution rules are irrelevant in the analysis, and accurate knowledge of the system can be realized with partially known information [34].

The gray correlation analysis procedure is described in detail below. IBM SPSS Statistics (version 24.0) was used for the calculation.

**Step 1:** Determination of the reference sequence

Let $X_{0}$ = {$X_{0}$(k), *k* = 1, 2, …, *m*} be the original reference sequence that reflects THE in China in 2000–2018, and let $X_{i}$ = {$X_{i}$(k), *i* = 1, 2, …, *n*} be the original comparative sequence that reflects the driving factors, such as economy, population, health service utilization, and policies.

**Step 2:** Initialization process

First, the original sequence is interpreted by dimensionless processing to avoid the effect of unit inconsistency on the correlation analysis; this paper uses the mean value processing approach. The sequences processed by initialization are denoted as $x^{\prime }(k)$ and expressed as shown in Equations (1) and (2).
(1)$${x_{i}}^{\prime }(k)=\frac{x_{i}\left(k\right)}{\bar{x_{i}}}$$
(2)$$\bar{x_{i}}=\frac{1}{m}\sum x_{i}\left(k\right)$$

**Step 3:** Calculation of the gray correlation coefficients of each sequence

The calculation method of the gray correlation coefficient is shown in Equation (3), where *ξ* is the resolution coefficient (within the [0,1] interval; the value is usually 0.5), $\Delta_{max}$ is the maximum difference between two sequences, and $\Delta_{min}$ is the minimum difference.
(3)$$\varepsilon_{i}(k)=\frac{\Delta_{min}+\xi *\Delta_{max}}{\Delta_{i}\left(k\right)+\xi *\Delta_{max})}$$

**Step 4:** Determination of correlation grade

Finally, the value of the correlation degree is $\beta_{i}$, which is shown as Equation (4), and the rank of the correlation degree among the driving factors is $\gamma_{i}$.
(4)$$\beta_{i}=\frac{1}{m}\sum \varepsilon_{i}\left(k\right)$$

### 3.2. Model of Gray Prediction

#### 3.2.1. Traditional Gray Model

The theory of the gray system is that all random quantities are gray quantities and gray processes that vary within a certain range and a certain period of time, and that no matter how complex the objective system is, it is always related, has overall functions and is therefore orderly. Therefore, when the gray system processes data, it is not seeking their statistical law and probability distribution, but rather seeks to make them into more regular time series data after processing them in a certain way, namely, as a “module”, and then builds a model. The module’s geometric meaning refers to the general term of the continuous curve and its bottom (i.e., abscissa) given on the two-dimensional plane of time and data. A module composed of known data columns is called a white module, and a module that is extrapolated from the white module to the future, that is, a module composed of predicted values, is called a gray module. Specifically, the module seeks to find the inherent laws in the irregular original data through the gray generation function and the differential fitting method in the case of poor information. Additionally, the module requires a small number of experimental data (at least four) for accurate prediction and has low data distribution requirements [8]. The traditional gray models are divided into two types, namely, GM(1,1) and GM(1,N). GM(1,1) is a univariate prediction model, and it does not consider which factors will influence the development of the system [35,36,37]. GM(1,N) represents the first-order gray model that has *N* variables, including the total number of (*N* − 1) independent variables and one dependent variable.

Suppose that there are a total of n variables denoted by $X_{i}{}^{\left(0\right)}$ and that each variable has m original sequences, as presented in Equation (5).
(5)$$X_{i}{}^{\left(0\right)}\left(k\right)=\left\{x_{i}{}^{\left(0\right)}\left(1\right),x_{i}{}^{\left(0\right)}\left(2\right)\ldots ,x_{i}{}^{\left(0\right)}\left(m\right)\right\}\;(i=\;1,\;2,\;\ldots ,\;n;\;k=\;1,\;2,\;\ldots ,\;m)$$

**Step 1:** Accumulated generating operation (1-AGO).

First, the original sequences of each variable can be processed by using 1-AGO, and $X_{i}{}^{\left(1\right)}$ is the 1st-order AGO sequence of $X_{i}{}^{\left(0\right)}$. The method of 1-AGO and $X_{i}{}^{\left(1\right)}$ is shown in Equations (6) and (7).
(6)$$x_{i}^{\left(1\right)}\left(k\right)=\overset{k}{\underset{1}{\sum }}x_{i}^{\left(0\right)}\left(k\right)$$
(7)$$x_{i}^{\left(1\right)}=\left\{x_{i}^{\left(1\right)}\left(1\right),x_{i}^{\left(1\right)}\left(2\right),\cdots ,x_{i}^{\left(1\right)}\left(m\right)\right\}$$

**Step 2:** Determining the driving parameters.

Equation (8) is the whitening differential equation of the GM(1,N) model.
(8)$$\frac{dx_{1}^{\left(1\right)}\left(k\right)}{dt}+ax_{1}^{\left(1\right)}\left(k\right)=\sum_{2}^{n}b_{i-1}x_{i}^{\left(1\right)}\left(k\right)\;(k=\;2,\;3,\;\ldots ,\;m)\;$$

Then, the gray differential equation can be obtained, as presented in Equation (9).
(9)$$x_{1}^{\left(0\right)}\left(k\right)+az_{1}^{\left(1\right)}\left(k\right)=\overset{n}{\underset{2}{\sum }}b_{i-1}x_{i}^{\left(1\right)}\left(k\right)$$
where $z_{1}^{\left(1\right)}\left(k\right)$ is defined as shown in Equation (10)
(10)$$z_{1}^{\left(1\right)}\left(k\right)=\frac{1}{2}\left[x_{1}^{\left(1\right)}\left(k\right)+x_{1}^{\left(1\right)}\left(k-1\right)\right]\;(k=\;2,\;3,\;\ldots ,\;m)$$
where *a* represents the system development parameter and $b_{i}$ represents the driving parameter.

Then, *Y, B*, and β are defined as shown in Equation (11), where Y=B∗β.
(11)$$\begin{array}{cc} Y=\left[\begin{array}{c} x_{1}^{\left(0\right)}\left(2\right)\\ x_{1}^{\left(0\right)}\left(3\right)\\ \vdots \\ x_{1}^{\left(0\right)}\left(m\right) \end{array}\right] & B=\left[\begin{array}{cccc} z_{2}^{\left(1\right)}\left(2\right) & x_{2}^{\left(1\right)}\left(2\right) & \cdots & x_{n}^{\left(1\right)}\left(2\right)\\ z_{2}^{\left(1\right)}\left(3\right) & x_{2}^{\left(1\right)}\left(3\right) & \cdots & x_{n}^{\left(1\right)}\left(3\right)\\ \vdots & \vdots & \vdots & \vdots \\ z_{2}^{\left(1\right)}\left(m\right) & x_{2}^{\left(1\right)}\left(m\right) & \cdots & x_{n}^{\left(1\right)}\left(m\right) \end{array}\right]\\ \beta =\left[\begin{array}{c} a\\ b_{1}\\ b_{2}\\ \vdots \\ b_{n-1} \end{array}\right] & \end{array}$$

In the GM(1,N) models, *Y* and *B* are known quantities, and *β* is the pending parameter. The gray parameter, $P_{N}$, represents the vector composed of the system development parameter, and the driving parameters can be obtained according to the least-squares method according to Equation (12).
(12)$$\overset{\wedge }{\beta }={\left(B^{T}B\right)}^{-1}B^{T}Y=\left[\begin{array}{c} \overset{\wedge }{a}\\ \overset{\wedge }{b_{1}}\\ \vdots \\ \overset{\wedge }{b_{n-1}} \end{array}\right]$$

**Step 3:** Prediction by using the inverse accumulated generating operation.

Then, the solution of the equation can be obtained by substituting the gray parameter in Equation (8), as presented in Equation (13), which is called the time-corresponding formula of GM(1,N).
(13)$$\overset{\wedge }{x_{1}^{\left(1\right)}}\left(k+1\right)=\left[x_{1}^{\left(0\right)}\left(1\right)-\frac{1}{\overset{\wedge }{a}}\overset{n}{\underset{i=2}{\sum }}\overset{\wedge }{\overset{\wedge }{b_{i-1}}x_{i}^{\left(1\right)}\left(k+1\right)}\right]e^{-\overset{\wedge }{a}k}+\frac{1}{\overset{\wedge }{a}}\overset{n}{\underset{i=2}{\sum }}\overset{\wedge }{b_{i-1}}x_{i}^{\left(1\right)}\left(k+1\right)$$

Finally, the *k* + 1-th predictive value can be obtained,$\;\overset{\wedge }{x_{1}^{\left(0\right)}}\left(k+1\right)$, through the inverse accumulated generating operation, as presented in Equation (14), which is called the accumulative subtraction formula of GM(1,N).
(14)$$\overset{\wedge }{x_{1}^{\left(0\right)}}\left(k+1\right)=\overset{\wedge }{x_{1}^{\left(1\right)}}\left(k+1\right)-\overset{\wedge }{x_{1}^{\left(1\right)}}\left(k\right)$$

#### 3.2.2. New Structure of the Multivariate Gray Prediction Model

The premise of the GM(1,N) model is fairly good because the system is whitened by many effective messages around its forecast origin. However, many scholars have noted some flaws in the existing GM(1,N) model’s prediction ability [38,39,40]. Zeng et al. [41], experts in gray prediction theory, pointed out three major defects in the traditional multivariate gray prediction model GM(1,N), that is, the mechanism’s defects caused by the over-idealization of the derivation process, the parameter’s defects caused by the “nonhomology” of parameter estimation and the application object, and the structural defects of lack of data mining and equivalent substitution. These are all important issues that affect the accuracy of the prediction model. They revised the GM(1,N) model in view of the defects and proposed a new structure of the multivariate gray prediction model, namely, NSGM(1,N), and the calculation method is as follows. The formulas and methods that are the same as those in GM(1,N) will not be repeated.

**Step 1:** Definition of the NSGM(1,N) model.

Consistent with the traditional gray prediction model GM(1,N), $X_{i}{}^{\left(0\right)}\left(k\right),x_{i}^{\left(1\right)}\left(k\right)$ and $z_{1}^{\left(1\right)}\left(k\right)$ are defined in the same way, but the model definition of NSGM(1,N) is different, as shown below in Equation (15).
(15)$$x_{1}^{\left(0\right)}\left(k\right)+az_{1}^{\left(1\right)}\left(k\right)=\sum_{2}^{n}b_{i}x_{i}^{\left(1\right)}\left(k\right)+h_{1}(k-1)+h_{2}$$

It is defined as a new gray model structure with a first-order equation and multiple variables, referred to as NSGM(1,N). The formula also contains system development parameters (*a*) and driving parameters (*b_i_*). At the same time, *h*_1_(*k* − 1) is defined as the linear correction term of the model, *h*_2_ is defined as the gray action, and the parameter is listed as $\overset{\wedge }{p}$ = [*b*_2_, *b*_3_, *b*_4_, ⋯, *b_n_*, *a*, *h*_1_, *h*_2_]. Therefore, the first-order model is shown as Equation (16).
(16)$$\overset{\wedge }{x_{1}^{\left(0\right)}}\left(k\right)=\;\sum_{2}^{n}b_{i}\overset{\wedge }{x_{1}^{\left(1\right)}}\left(k\right)-az_{1}^{\left(1\right)}\left(k\right)+h_{1}(k-1)+h_{2}$$

**Step 2:** Parameter estimation of the NSGM(1,N) model.

The least-squares method was also used to solve the parameter $\overset{\wedge }{p}$ in the NSGM(1,N) model, as shown in Equation (17) and Equation (18).
(17)$$\overset{\wedge }{p}={\left(B^{T}B\right)}^{-1}B^{T}Y$$
(18)$$Y=\left[\begin{array}{c} x_{1}^{\left(0\right)}\left(2\right)\\ x_{1}^{\left(0\right)}\left(3\right)\\ \vdots \\ x_{1}^{\left(0\right)}\left(m\right) \end{array}\right]B=\left[\begin{array}{cccc} x_{2}^{\left(1\right)}\left(2\right)\;x_{3}^{\left(1\right)}\left(2\right)\;\cdots \;x_{N}^{\left(1\right)}\left(2\right) & -z_{1}^{\left(1\right)}\left(2\right) & 1 & 1\\ x_{2}^{\left(1\right)}\left(3\right)\;x_{3}^{\left(1\right)}\left(3\right)\;\cdots \;x_{N}^{\left(1\right)}\left(3\right) & x_{2}^{\left(1\right)}\left(3\right) & 2 & 1\\ \vdots & \vdots & \vdots & \vdots \\ x_{2}^{\left(1\right)}\left(m\right)\;x_{3}^{\left(1\right)}\left(m\right)\;\cdots \;x_{N}^{\left(1\right)}\left(m\right) & x_{2}^{\left(1\right)}\left(m\right) & m-1 & 1 \end{array}\right]$$

**Step 3:** Time-corresponding formula and accumulative subtraction formula of the NSGM(1,N) model.

Equation (8) is used to derive the time-response formula in the GM(1,N) model. However, the parameters estimated by Equation (9) are used as the parameters of the time-response function, which leads to the “nonhomology” of parameter estimation and the application object. The NSGM(1,N) model uses one first-order equation, which is an equivalent modification of Equation (15), to derive the time response of NSGM(1,N), which ensures parameter estimation “homology” with parameter application. Therefore, the time-response formula is shown as Equation (19).
(19)$$\overset{\wedge }{x_{1}^{\left(1\right)}}\left(k\right)=\sum_{t=1}^{k-1}\left[\mu_{1}\sum_{i=2}^{N}\mu_{2}^{t-1}b_{i}x_{i}^{\left(1\right)}\left(k-t+1\right)\right]+\;\mu_{2}^{k-1}\overset{\wedge }{x_{1}^{\left(1\right)}}\left(1\right)\;+\;\sum_{j=0}^{k-2}\left[\mu_{2}^{j}\left(k-j\right)\mu_{3}+\mu_{4}\right]\;(k=2,3,4,\ldots ,m)$$

The accumulative subtraction formula of NSGM(1,N) is shown as Equation (20).
(20)$$\overset{\wedge }{x_{1}^{\left(0\right)}}\left(k\right)=\mu_{1}(\mu_{2}-1)\sum_{t=1}^{k-2}\left[\sum_{i=2}^{N}\mu_{2}^{t-1}b_{i}x_{i}^{\left(1\right)}\left(k-t\right)\right]+\;\mu_{1}\sum_{i=2}^{N}b_{i}x_{i}^{1}\left(k\right)+\;\sum_{j=0}^{k-3}\mu_{2}^{j}\mu_{3}+\left(\mu_{2}-1\right)\mu_{2}^{k-2}x_{1}^{\left(1\right)}\left(1\right)\;+\mu_{2}^{k-2}\left(2\mu_{3}+\mu_{4}\right)\;(k=2,\;3,\;4,\;\ldots ,\;m)$$
where
$$\mu_{1}=\frac{1}{1+0.5a}\;\mu_{2}=\frac{1-0.5a}{1+0.5a}\;\mu_{3}=\frac{h_{1}}{1+0.5a}\;\mu_{4}=\frac{h_{2}-h_{1}}{1+0.5a}.$$

## 4. Results

### 4.1. Description of Total Health Expenditure and Main Driving Factors

The description of THE and the main driving factors for 2000–2018 are shown in Table 1 and Figure 1. In China in 2000, THE was 458.66 billion yuan, accounting for 4.57% of GDP, and by 2018, THE was 5912.19 billion yuan, accounting for 6.57% of GDP. From 2000 to 2018, THE increased by 1189.01%, with an average annual growth rate of 15.26%, much higher than the GDP annual average growth rate of 12.97% in the same period. At the same time, GGHE grew rapidly, with the proportion of THE rising from 38.38% to 53.82%, while the OOP proportion of THE gradually decreased from 58.98% to 28.61%, which was very close to the target of 28% set in China’s 13th five-year (2016–2020) health and wellness plan. Moreover, residents’ consumption and living standards were constantly improving, and HCE was increasing from year to year, with an average annual growth of 11.16%. In terms of demographic factors, China’s population was gradually increasing, but the natural population growth rate was decreasing. ABOVE65 showed an upward trend that reached 11.94% in 2018, indicating that the aging of Chinese society was serious. In terms of health institutions, PER and BED were constantly increasing; in 2018, they increased by 112.19% and 164.53% compared with 2000, and the average annual growth rates were 4.27% and 5.55%, respectively.

### 4.2. Results of the Main Driving Factor Analysis

The degree of correlation between GGHE and the change in THE was 0.941, ranking first among all the factors, suggesting that GGHE was the factor most closely related to the change in THE. This finding indicated that on the one hand, the government plays a leading role in the health industry and has a critical impact on the development of health services, and on the other hand, through the implementation and improvement of social insurance policies, the government has enabled social medical insurance funds to play a vital role in ensuring the health of the population. Among other health policy factors, the degree of correlation between OOP and THE was 0.878, ranking fourth, suggesting that changes in the proportion of residents’ OOP will also have a great impact on changes in THE.

The degrees of correlation between GDP and THE and between HCE and THE were 0.910 and 0.904, and the correlation grades second and third, respectively, indicated that the development of China’s economy and the increase in residents’ income are closely related to improvements in health.

Additionally, the results show that the degrees of correlation between BED and THE and between PRE and THE were 0.791 and 0.756, respectively, indicating that the development of the economy has caused the development of health institutions, and that the availability of health services is also increasing, which promotes the provision and utilization of healthcare services and has an important impact on THE.

Additionally, the correlation degree between ABOVE65 and THE was 0.723, proving that the aging of the population was an important driving factor affecting THE, whereas the correlation degree between POP and THE was 0.672.

Last, the degree of correlation between INF and THE was below 0.6, at only 0.573, indicating that INF had less impact on THE than other factors.

### 4.3. Prediction Model of Total Health Expenditure

To evaluate the prediction accuracy of the model, all the experimental data were divided into two parts: training (2000–2016) and test data (2017 and 2018). The training data were used for the training of the model, and the test data were used to evaluate the predictive potential of the NSGM(1,10) model.

First, all 10 variables were included in the model to establish NSGM(1,10), including THE as the dependent variable and the 9 driving factors as independent variables.

When the MATLAB processing codes of NSGM(1,10) are run, the gray parameter can be obtained as shown in Equation (21).
(21)$$\begin{array}{c} \overset{\wedge }{\beta }={\left(b_{2},b_{3},b_{4},b_{5},b_{6},b_{7},b_{8},b_{9},b_{10},a,h_{1},h_{2}\right)}^{T}\;=\;(-10.4199,\;17.9940,\;0.0430,\;0.0428,\;0.5213,\;1.9117,\;0.5433,\;0.1432,\\ \;22.3902,\;0.6942,\;21,536.5233,\;20,977.0284) \end{array}$$

Table 2 shows the results of using NSGM(1,10) to compare the prediction of THE with the actual data. The residual percentage of the training data (2000–2016) is within 1%, except for 2004 and 2007, where it is slightly higher than 1%, and the average residual percentage of the training data is 0.36%. Additionally, the residual percentages of prediction and actual data of the test data (2017–2018) are 1.36% and 2.34%, respectively, and the average residual percentage is only 1.85%. Thus, the NSGM(1,10) model has good fit and predictive ability.

To verify the superiority of the prediction results of the NSGM(1,10) model, this paper also used the traditional gray prediction models, GM(1,1) and GM(1,N), to fit and predict the data.

When the MATLAB processing codes of GM(1,1) are run, the gray parameter is calculated as a = −0.147532 and b = 430.168721. The results shown in Table 2 indicate that the residual percentage in 2004–2007 is more than 10%, while the residual percentage in 2009 and 2013 is within 1%, and the average residual percentage of the training data is 6.06%. Thus, the fit of GM(1,1) for the training data (2000–2016) is poor and unstable. Moreover, the residual percentages of the prediction and actual data of the test data (2017–2018) are 8.07% and 11.43%, respectively, and the average residual percentage is 9.75%, which is close to 10%, so there is a large gap between the prediction and the actual data.

Then, by following the procedures of GM(1,10), the gray parameter was calculated as
(22)$$\overset{\wedge }{\beta }={\left(a,b_{1},b_{2},b_{3},b_{4},b_{5},b_{6},b_{7},b_{8},b_{9}\right)}^{T}\;=\;(0.9170,\;85.818,\;80.035,\;0.011,\;0.009,0.506,\;0.963,\;2.522,\;0.570,\;8.825)$$

Table 2 shows the results of using GM(1,10) to compare the prediction of THE with the actual data. The residual percentages of the training data (2000–2016) are relatively large, even exceeding 20% in 2002–2003, and the average residual percentage of the training data is 10.97%, which is the largest among the three methods. For the test data (2017–2018), the residual percentages of the prediction and actual data are 10.97% and 6.82%, respectively, and the average residual percentage is 6.82%, which is better than the test results of GM(1,1).

Finally, to show the advantage of the gray model for predicting THE with scant data, we also used the BP neural network, which is widely used in the field of prediction, to predict THE. As in NSGM(1,N), we used 2017–2018 data as test data to evaluate the prediction accuracy of the model. MATLAB was used to perform the BP neural network model, and the final model contained a two-layer feedforward network. There were six TANSIG hidden neurons in the hidden layer and one PRRELIN neuron in the output layer, and the TRAINLM network training function was used.

Finally, because of the uncertainty of the results of the BP neural network model, we selected the 10 best results with the smallest residual percentage, and the residual percentages of the 10 predictions are listed in Table 3. The results show that there are great differences among the 10 results. The minimum residual percentage of the training data is 0.245%, and the maximum is 5.628%. At the same time, the minimum residual percentage of the test data is 1.504%, and the maximum is 9.093%. The average residual percentages of the 10 results of the training data and the test data are 1.140% and 2.930%, respectively, and we also reach a good level of fit and prediction. Although the minimum residual percentage of the training data and test data is smaller than in the results of NSGM(1,10), the results of NSGM(1,10) are better than the 10-time average of the BP neural network model.

The comparison among the predictions of the four predictive methods and the actual data is shown in Figure 2, which indicates that the curve of NSGM(1,N) is closest to the actual data.

## 5. Discussion

This paper uses data from 2000–2018, and selects nine hot-topic factors in health areas to analyze the main driving factors of the growth in THE and to establish and test the THE prediction model. The nine hot topic factors have good representativeness and largely reflect the trend of the THE. At the same time, the results of the prediction model are excellent, and the predicted models are excellent.

In public policy and health institutions, the government and society have played a positive role in improving health conditions and reducing the economic burden. When the government proposed a new health system reform in China in 2009, it needed to increase the input of government in all fields, accelerate the establishment and improvement of a multilevel medical security system covering urban and rural residents, and improve the primary health service system to promote fairness and efficiency in the medical and health industry [16]. We can find that through the formulation of policies, the government and society have invested heavily in healthcare and health institution construction, and this investment has played a vital role in improving the fairness and efficiency of the healthcare system. Therefore, the proportion of OOP has shown a downward trend year by year, which is gradually approaching the goal set by China’s 13th five-year (2016–2020) health and wellness plan (28%), whereas the OOP health expenditure is still increasing rapidly, and the medical burden of residents has not been substantially alleviated. Therefore, improving the allocation efficiency of government health expenditure, perfecting the medical insurance system [42] and increasing society’s role in sharing the risk of heath expenditure are valuable in controlling the growth of THE and reducing residents’ medical burden.

In terms of demographics and the economy, aging is a major concern. According to the current report [43], the population is aging rapidly as a result of the baby boom, the One Child Policy and the declining mortality rate, and the demographic household structure is gradually becoming a “4–2–1” or “4–2–2” formula, meaning the elderly population will continue to increase. Moreover, this paper demonstrates that aging is closely related to the growth in total health expenditure, which is consistent with the present study [38,39,40]. Therefore, this paper proposes that aging provides more opportunities for the increase in THE and is a carrier that can combine the improvement of the economy, medical insurance, and medical science with the health of the population, and convert them to health expenditure. Therefore, in the context of the increasing number of older people, there is an urgent need to pay more attention to the health of the elderly, develop strategies for preventive and rehabilitative care, particularly in the medical treatment and nursing of the elderly, and formulate corresponding insurance strategies to reduce the medical burden of the elderly. Moreover, according to the present study [44], health literacy is inversely proportional to the utilization of and expenditure on healthcare. Therefore, it is necessary to adopt health education and knowledge popularization measures to improve the health literacy of the population, including elderly and young people, to control health expenditure. At the same time, measures such as providing regular physical examinations and improving sanitation facilities for the population can be used to transform the fruits of economic development into health improvements for elderly and young people.

Compared with the traditional gray prediction models, GM(1,1) and GM(1,N), the improved NSGM(1,N) model not only avoids the problems of the GM(1,N) model but also improves the predictive accuracy. The predictability of the NSGM(1,N) model is better than that of the BP neural network; besides, the predictions of the BP neural network were different and varied greatly in each run of the code, whereas the predictions of NSGM(1,N) were certain when the training data were determined, so NSGM(1,N) has better prediction stability. Finally, the BP neural network model is suitable for the prediction of more data. Considering that the model was established in terms of fewer data, the predicted simulation sequence for the training data was very close to the original sequence, and the residual percentage is small, we conclude that the improved NSGM(1,N) model can predict health expenditure fairly accurately in the situation of poor information, with results superior to those of the traditional gray model and BP neural network model. Therefore, the NSGM(1,N) gray prediction model has good applicability in predicting THE.

In terms of practical implications, on the one hand, this paper innovatively introduces grey system theory into the field of health, proving its application value in health expenditure prediction. On the other hand, the analysis of the main driving factors of THE enables us to grasp the reasons leading to the growth in THE, which is crucial for the formulation of effective, strategic health service policies to facilitate the progress of China’s new health system reform.

Our analysis has three main limitations. First, although the nine factors we selected were well-represented and the predictive model was accurate, they were limited and could not fully explain the increase in total health expenditure. Second, there are great differences in the economies, populations and policies of different regions in China, and this article can only reflect the overall situation in China rather than the situation in a certain region. Third, although the nine variables are well represented and the prediction results are good, there are correlations among some variables, which may affect the prediction performance of the GM model to a certain extent.

## 6. Conclusions

Given this study’s analysis, the following conclusions can be drawn. First, under the socialist system, the policies and investment of the Chinese government and society have played a crucial role in reducing the burden on people. In addition, China’s medical system reform has been effective, and the proportion of OOP has gradually decreased from year to year. To a certain extent, residents’ medical burden has been reduced. Second, the improvement of the economy and the aging of the population, which are closely related to THE, have increased the demand for health services, leading to continuous increases in THE, so improving the efficiency of investment and providing preventive healthcare and nursing for the elderly are crucial. Third, the improved NSGM(1,N) model achieves good prediction accuracy as it has unique advantages in simulating and predicting THE; thus, it can provide a basis for policy formulation. In the future research, we will focus on health policies and the aging of the population to analyze the changes in health expenditure, and then explore measures to control the unreasonable increase in medical expenditure. Besides this, it is also an important direction to analyze and compare the prediction effects of various multivariate grey models.

## Figures and Tables

**Figure 1 healthcare-09-00207-f001:**
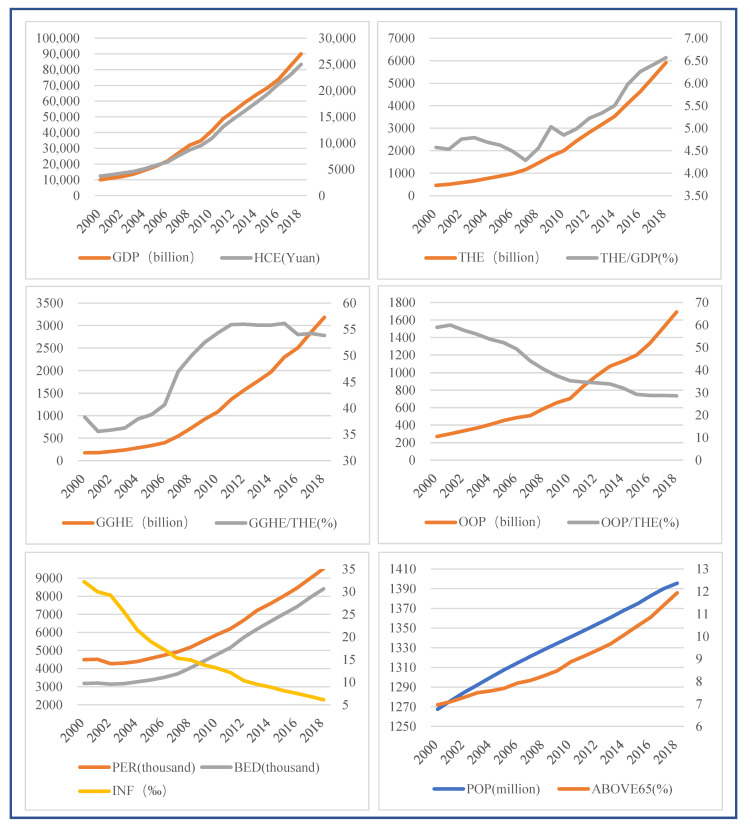
Description of total health expenditure and main driving factors.

**Figure 2 healthcare-09-00207-f002:**
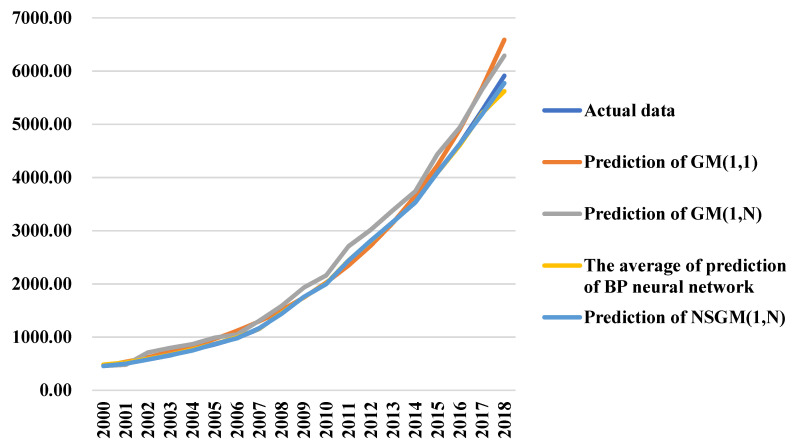
The similarity between the predictions and the actual data.

**Table 1 healthcare-09-00207-t001:** The description of THE and main driving factors for 2000–2018.

YEAR	THE ^a^	ABOVE65 ^b^	POP ^c^	GDP ^d^	PER ^e^	BED ^f^	GGHE ^g^	OOP ^h^	INF ^i^	HCE ^j^
2000	458.66	88.21	1267.43	10,028.01	4490.80	3177.00	175.591	270.517	32.2	3721
2001	502.59	90.62	1276.27	11,086.31	4508.00	3201.00	178.78	301.388	30	3987
2002	579.00	93.77	1284.53	12,171.74	4270.00	3136.00	207.478	334.214	29.2	4301
2003	658.41	96.92	1292.27	13,742.20	4381.00	3164.00	238.514	367.867	25.5	4606
2004	759.03	98.57	1299.88	16,184.02	4486.00	3268.00	288.23	407.135	21.5	5138
2005	865.99	100.55	1307.56	18,731.89	4564.05	3367.50	335.713	452.097	19	5771
2006	984.33	104.19	1314.48	21,943.85	4728.35	3511.80	400.17	485.356	17.2	6416
2007	1157.40	106.36	1321.29	27,009.23	4913.19	3701.10	543.166	509.866	15.3	7572
2008	1453.54	109.56	1328.02	31,924.46	5174.48	4038.70	726.035	587.586	14.9	8707
2009	1754.19	113.07	1334.50	34,851.77	5535.12	4416.60	920.92	657.116	13.80	9514.00
2010	1998.04	118.94	1340.91	41,211.93	5876.16	4786.80	1085.16	705.129	13.10	10,919.00
2011	2434.59	122.88	1347.35	48,794.02	6202.86	5159.90	1360.70	846.528	12.10	13,134.00
2012	2811.90	127.14	1354.04	53,858.00	6675.55	5724.80	1573.43	965.632	10.30	14,699.00
2013	3166.90	131.61	1360.72	59,296.32	7210.58	6181.90	1767.35	1072.934	9.50	16,190.00
2014	3531.24	137.55	1367.82	64,128.06	7589.79	6601.20	1969.96	1129.541	8.90	17,778.00
2015	4097.46	143.86	1374.62	68,599.29	8007.54	7015.20	2299.99	1199.265	8.10	19,397.00
2016	4634.49	150.03	1382.71	74,006.08	8454.40	7410.50	2502.69	1333.79	7.50	21,285.00
2017	5259.83	158.31	1390.08	82,075.43	8988.23	7940.30	2850.49	1513.36	6.80	22,935.00
2018	5912.19	166.58	1395.38	90,030.95	9529.18	8404.10	3182.16	1691.20	6.10	25,002.00

^a^ Total health expenditure (billion). ^b^ Number of people aged 65 and over (million). ^c^ Population (million). ^d^ Gross domestic product (billion). ^e^ Number of medical technical personnel (thousand). ^f^ Number of beds in healthcare institutions (thousand). ^g^ General government expenditure on health (billion). ^h^ Out-of-pocket health expenditure (billion)^. i^ Infant mortality rate (‰). ^j^ Household consumption expenditure (Yuan).

**Table 2 healthcare-09-00207-t002:** Comparison of the actual data and prediction by the gray prediction model.

Year	Actual Data	GM(1,1)		GM(1,10)		NSGM(1,10)	
		Prediction	${\phi_{i}}^{d}$	Prediction	$\phi_{i}$	Prediction	$\phi_{i}$
2000	458.66	458.66	-	458.66	-	458.66	-
2001	502.59	536.43	6.73	481.02	4.29	502.23	0.07
2002	579.00	621.71	7.38	713.56	23.24	578.52	0.08
2003	658.41	720.55	9.44	797.19	21.08	663.27	0.74
2004	759.03	835.09	10.02	869.58	14.56	751.09	1.05
2005	865.99	967.85	11.76	990.47	14.37	871.22	0.60
2006	984.33	1121.70	13.96	1050.55	6.73	979.24	0.52
2007	1157.40	1300.00	12.32	1308.47	13.05	1170.07	1.09
2008	1453.54	1506.70	3.66	1587.91	9.24	1439.62	0.96
2009	1754.19	1746.20	0.46	1932.70	10.18	1759.01	0.27
2010	1998.04	2023.80	1.29	2159.12	8.06	1996.37	0.08
2011	2434.59	2345.50	3.66	2707.82	11.22	2437.18	0.11
2012	2811.90	2718.40	3.33	3017.02	7.29	2812.44	0.02
2013	3166.90	3150.60	0.51	3384.96	6.89	3164.62	0.07
2014	3531.24	3651.40	3.40	3740.85	5.94	3533.78	0.07
2015	4097.46	4231.90	3.28	4441.08	8.39	4095.74	0.04
2016	4634.49	4904.60	5.83	4942.22	6.64	4634.52	0.00
**RE_1_ (%) ^a^**			**6.06**		**10.97**		**0.36**
2017 ^b^	5259.83	5684.3	8.07	5650.81	7.43	5188.14	1.36
2018 ^b^	5912.19	6587.9	11.43	6289.61	6.38	5774.04	2.34
**RE_2_(%) ^c^**			**9.75**		**6.82**		**1.85**

^a^ The average residual percentage (RE) of training data. ^b^ Data used for testing. ^c^ The average residual percentage of test data. ^d^
$\phi_{i}=\frac{\left|\mathrm{Actual}\;\mathrm{data}-\mathrm{Prediction}\right|}{\mathrm{Actual}\;\mathrm{data}}$ × 100%.

**Table 3 healthcare-09-00207-t003:** Residual percentage of 10 predictions by the BP neural network (%).

Year	Time										Mean ^d^
	1	2	3	4	5	6	7	8	9	10	
2000	13.204	0.686	19.718	1.357	4.493	1.275	6.789	0.428	7.081	13.204	5.447
2001	8.942	1.216	17.543	0.816	1.578	0.549	6.228	0.224	3.232	8.942	4.259
2002	6.059	0.360	15.047	0.750	2.074	0.100	3.195	0.150	3.236	6.059	2.519
2003	2.757	0.780	11.839	0.290	1.332	0.053	4.390	0.133	1.651	2.757	1.808
2004	0.780	0.426	9.497	0.127	1.650	0.434	4.090	0.291	0.021	0.780	1.282
2005	0.150	0.007	7.105	0.127	1.871	0.420	3.557	0.161	0.674	0.150	0.744
2006	0.224	0.113	4.757	0.020	0.475	0.327	5.138	0.029	1.071	0.224	0.984
2007	0.412	0.971	2.742	0.202	0.878	0.329	6.645	0.227	3.584	0.412	0.432
2008	0.437	0.035	0.866	0.017	1.039	0.021	1.626	0.487	0.009	0.437	0.016
2009	0.382	0.048	0.304	0.090	0.727	0.464	0.182	0.482	1.968	0.382	0.048
2010	0.133	0.232	1.227	0.079	0.682	0.868	1.850	0.291	3.385	0.133	0.395
2011	0.096	0.086	1.362	0.031	0.305	1.073	1.628	0.442	0.163	0.096	0.195
2012	0.169	0.063	1.311	0.069	0.621	1.321	1.251	0.474	2.980	0.169	0.152
2013	0.210	0.804	0.929	0.093	0.800	1.262	2.189	0.242	2.632	0.210	0.144
2014	0.234	0.728	0.645	0.057	1.665	1.003	2.289	0.922	1.948	0.234	0.180
2015	0.181	1.351	0.454	0.008	2.819	0.393	2.300	1.195	1.095	0.181	0.166
2016	0.150	1.176	0.335	0.036	2.902	0.078	1.115	0.916	2.212	0.150	0.608
**RE_1_^a^**	**2.031**	**0.534**	**5.628**	**0.245**	**1.524**	**0.586**	**3.204**	**0.417**	**2.173**	**2.031**	**1.140**
2017 ^b^	3.107	0.524	1.505	1.377	5.847	2.816	1.578	3.370	8.322	3.107	4.890
2018 ^b^	1.301	3.324	7.346	1.632	12.339	4.132	6.046	9.962	16.432	1.301	0.969
**RE_2_^c^**	**2.204**	**1.924**	**4.426**	**1.504**	**9.093**	**3.474**	**3.812**	**6.666**	**12.377**	**2.204**	**2.930**
MSE	0.00010	0.00010	0.00063	0.00000	0.00051	0.00008	0.00056	0.00008	0.00055	0.00010	-

^a^ The average of the residual percentage of training data. ^b^ Data used for testing. ^c^ The average of the residual percentage of test data. ^d^ The residual percentage of the mean of the 10 predictions.

## Data Availability

The datasets used and analyzed during the current study are available from the corresponding author on reasonable request.

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
