# Peer review of "Total Health Expenditure and Its Driving Factors in China: A Gray Theory Analysis"

_healthcare, 2021, doi:10.3390/healthcare9020207_

Round 1

Reviewer 1 Report

The authors of the research paper “Total health expenditure and its driving factors in China: a gray theory analysis “presented a topic very relevant in its field

Honestly I liked this paper very much.  The paper is very well presented and structured and is very easy and pleasant to read providing plenty of information about the topic.

I consider that methodology and empirical results have been properly realized and the research methodology is very appropriate.

Some minor comments to improve the quality of this research paper.

The point 3.2 Economy and 3.4. Health institutions should be enlarged and providing with much more detailed information.

I would strongly recommend the authors to include some limitations in this study

Finally, I would like to see also a much more detailed information about future research.

Perhaps it would be interesting to rewrite the last section of conclusions, including the limitations and future research in it.

Author Response

Dear Reviewer,

We appreciate for your constructive comments on our paper. We would like to submit the revised manuscript entitled “Total health expenditure and its driving factors in China: a gray theory analysis” (manuscript ID: healthcare-1098992). We have carefully revised the manuscript text based on your comments, and the "Track Changes" function was used to mark the changes.  If there are other questions about the paper, I hope you will not hesitate to raise them to help us improve the quality of my paper and publish in “Healthcare”. The following is the point-by-point responses to your comments.

Thank you again for your generosity and best regards.

Sincerely,

Xihe Yu

Point 1: The point 3.2 Economy and 3.4. Health institutions should be enlarged and providing with much more detailed information.

Response 1: We have revised the content in Economy and Health institutions and provided with much more detailed information. (p. 4-5)

Point 2: I would strongly recommend the authors to include some limitations in this study.

Response 2: We have revised the limitations of this study in the last part of the discussion. (p. 22, lines 746-755)

Point 3: I would like to see also a much more detailed information about future research.

Response 3: Thank you very much for your interest in our research. In the future research, on the one hand, we will continue to pay attention to the changes in the total health expenditure in China, and take health policies and the aging of the population as the research focus to analyze the changes, and then explore measures to control the unreasonable increase in medical expenditure. On the other hand, based on the traditional gray model, previous studies have proposed a variety of improved models. Therefore, in the next step, we will have a deep understanding of other multiple gray models, and analyze and compare the prediction effects of various models.

Point 4: Perhaps it would be interesting to rewrite the last section of conclusions, including the limitations and future research in it.

Response 4: We appreciate your suggestion. We have revised the limitations of the research and put them in the last part of the discussion in accordance with the requirements of the journal. Besides, we added future research directions in the last section of the conclusion. (p. 22, lines 770–774)

Reviewer 2 Report

Review Comment

Title: Total health expenditure and its driving factors in China: a gray theory analysis

Summary:

The paper conducted the analysis on total health expenditure and driving factors in China using gray model (GM). The multivariate gray model achieved good prediction accuracy in total health expenditure over time, especially when the amount of data is limited. The experiments and analysis results are sufficient and can be justified. After addressing a few concerns below, the paper should be good for publication.

General Comments:

  1. What are the assumptions on variables in GM model? Does the GM model need feature selection? Among the driving factors, several pairs of them can be highly correlated. For example, GDP and HCE, PER and BED. What will be the model performance after removing some of the highly-correlated features?
  2. Page 17, please elaborate more on the meanings of GM parameters and how informative they are for different decision making process such as government policy making, healthcare resource allocation, etc.

Specific Comments:

  1. Page 7, Lines 245-248, notations of parameters should be italic.
  2. Page 10. Line 396, ‘PRE’ should be ‘PER’
  3. Page 12, figure is out of the box.
  4. The shot of NSGM for Multivariate Gray Prediction Model is not intuitive. Maybe MGM (Multivariate Gray Model) is a better option.

Author Response

Dear Reviewer,

We appreciate for your constructive comments on our paper. We would like to submit the revised manuscript entitled “Total health expenditure and its driving factors in China: a gray theory analysis” (manuscript ID: healthcare-1098992). We have carefully revised the manuscript text based on your comments, and the "Track Changes" function was used to mark the changes.  If there are other questions about the paper, I hope you will not hesitate to raise them to help us improve the quality of my paper and publish in “Healthcare”. The following is the point-by-point responses to your comments.

Thank you again for your generosity and best regards.

Sincerely,

Xihe Yu

Point 1: What are the assumptions on variables in GM model? Does the GM model need feature selection? Among the driving factors, several pairs of them can be highly correlated. For example, GDP and HCE, PER and BED. What will be the model performance after removing some of the highly-correlated features?

Response 1: According to the grey system theory and previous researches, the GM model has no feature requirements for variables. We admit that there is a correlation between some driving variables, but the 9 variables involved in this paper are factors in previous research and hot topics in China, so we think they are well representative. At the same time, after we removed some highly-correlated variables, the prediction results were not as good as the current results. We appreciate your constructive comment, so we mentioned the collinearity of the independent variables in the limitations of the study and described it as " Third, although the 9 variables are well represented and the prediction results are good, there are correlations among some variables, which may affect the prediction performance of the GM model to a certain extent." (p. 22, lines 752–755)

Point 2: Page 17, please elaborate more on the meanings of GM parameters and how informative they are for different decision making process such as government policy making, healthcare resource allocation, etc.

Response 2: In GM, a represents the system development parameter which represents the development trend of the estimated value of the behavior sequence; and b represents the driving parameter, which is the data mined from the behavior sequence and reflects the relationship of data changes. In NSGM, a and b have the same interpretation as GM, and h is defined as the linear correction term of the model. We have consulted the grey system theory and previous researches, and proved that the magnitude and positive and negative values of these parameters cannot directly reflect the effect of each factors. They are only the values generated from the perspective of mathematical modeling when predicting, and have no practical significance. Besides, this paper used the Grey Correlation Analysis to reflect the degree of correlation between each factor and the total health expenditure. Therefore, we can’t respond to your comments perfectly, please forgive us.

Point 3: Page 7, Lines 245-248, notations of parameters should be italic.

Response 3: We have changed the format of notations to italic. (p. 9, lines 371-375)

Point 4: Page 10. Line 396, ‘PRE’ should be ‘PER’

Response 4: We have corrected "PRE" to "PER". (p. 10, lines 422)

Point 5: Page 12, figure is out of the box.

Response 5: We have revised the Fig 1. (p. 12)

Point 6: The shot of NSGM for Multivariate Gray Prediction Model is not intuitive. Maybe MGM (Multivariate Gray Model) is a better option.

Response 6: We appreciate your comments and suggestions very much. This paper introduced the NSGM model to the field of health economy, focusing on its application effects in this paper. We were inspired by your suggestion, so in the next step, we will learn MGM (Multivariate Gray Model) in depth and  compare the two models.

Reviewer 3 Report

Thank you for the opportunity to review this manuscript. Overall, this is an important topic where the objective was to analyze the main 11 driving factors of THE in China in the 21st century and establish a predictive 12 model, Gray system theory was employed to explore the correlation degree be-13 tween THE and 9 hot topics in the areas of the economy, population, health ser-14 vice utilization, and policy using national data from 2000 to 2018. There is good data in this study, however, here are some suggestions to improve the manuscript. Please see my comments just below, they are listed according to the article order:

- In the Abstract section, please consider explain this sentence: “Additionally, the NSGM(1,N) prediction model of health expenditure was established and compared with the traditional grey model, namely GM(1,1) and GM(1,N) and widely  used BP neural network”.

The introduction should provide sufficient background and include all relevant references about rationale for this study or a conceptual framework to help the reader understand the consideration of variables you have studied (9 hot-topic factors in health areas).

The methods are adequately described. The research design is appropriate.

Results
I find it hard to understand how interpret the data in Table 3. Please add an explanation and be more explicit in the manuscript as to what you are showing.

Discussion
- Start the discussion by reporting your own findings from the present study and then, after that, you put it in perspective of other available research.
- It is sometimes hard to follow the discussion as it is unclear when you talk about your own study and results or another study. Please read your text carefully and use phrases such as "in the present study…" or "the current report" or similar.
- Do not repeat your own results (numbers) in the discussion if not absolutely necessary to make a certain point. Use descriptive language instead.

- Please, write a practical implications section.

The conclusions are supported by the results.

Author Response

Dear Reviewer,

We appreciate for your constructive comments on our paper. We would like to submit the revised manuscript entitled “Total health expenditure and its driving factors in China: a gray theory analysis” (manuscript ID: healthcare-1098992). We have carefully revised the manuscript text based on your comments, and the "Track Changes" function was used to mark the changes.  If there are other questions about the paper, I hope you will not hesitate to raise them to help us improve the quality of my paper and publish in “Healthcare”. The following is the point-by-point responses to your comments.

Thank you again for your generosity and best regards.

Sincerely,

Xihe Yu

Point 1: In the Abstract section, please consider explain this sentence: “Additionally, the NSGM(1,N) prediction model of health expenditure was established and compared with the traditional grey model, namely GM(1,1) and GM(1,N) and widely  used BP neural network”.

Response 1: We have revised this sentence in the abstract to “Additionally, the NSGM(1,N) prediction model of total health expenditure was established and compared with the traditional grey model and widely used BP neural network to evaluate the prediction effectiveness of the model. (p. 1, lines 15-18)

Point 2: The introduction should provide sufficient background and include all relevant references about rationale for this study or a conceptual framework to help the reader understand the consideration of variables you have studied (9 hot-topic factors in health areas).

Response 2: We appreciate your constructive comments. Because the aim of this article is the analysis of China's total health expenditure and its driving factors, the introduction mainly introduces the current status of health expenditure in some countries, the direction and methods of previous research, and the total health expenditure in China. After considering your comments, we have changed the 3 Data Collection to 2 Materials and expanded the explanation of some variables so that readers can understand our consideration of variables.

Point 3: I find it hard to understand how interpret the data in Table 3. Please add an explanation and be more explicit in the manuscript as to what you are showing.

Response 3: The predictions of the BP neural network are different and varied greatly in each run of the code, which is a drawback of the BP neural network, and similar to the previous research, we chose the 10 best results. In order to show more clearly what is described in Table 3, we have revised the section. (p. 17, lines 606-608)

Point 4: Start the discussion by reporting your own findings from the present study and then, after that, you put it in perspective of other available research.

Response 4: We have restructured the discussion. The discussion is first to report my research findings, then presents the practical implications, and finally discusses the limitations of the research. (p. 20-22)

Point 5: It is sometimes hard to follow the discussion as it is unclear when you talk about your own study and results or another study. Please read your text carefully and use phrases such as "in the present study…" or "the current report" or similar.

Response 5: We appreciate your constructive comments. We have read my text carefully and added the phrases in the appropriate places. (p. 21, lines 686,692,701)

Point 6: Do not repeat your own results (numbers) in the discussion if not absolutely necessary to make a certain point. Use descriptive language instead.

Response 6: We have revised the discussion and deleted the content about the results.

Point 7: Please, write a practical implications section.

Response 7: We have added the practical implications section in 5 Discussion. (p. 21, lines 726-732)